# Dietary Inulin to Improve SARS-CoV-2 Vaccine Response in Kidney Transplant Recipients: The RIVASTIM-Inulin Randomised Controlled Trial

**DOI:** 10.3390/vaccines12060608

**Published:** 2024-06-03

**Authors:** Julian Singer, Matthew J. Tunbridge, Bree Shi, Griffith B. Perkins, Cheng Sheng Chai, Tania Salehi, Beatrice Z. Sim, Svjetlana Kireta, Julie K. Johnston, Anouschka Akerman, Vanessa Milogiannakis, Anupriya Aggarwal, Stuart Turville, Pravin Hissaria, Tracey Ying, Huiling Wu, Branka Grubor-Bauk, P. Toby Coates, Steven J. Chadban

**Affiliations:** 1Department of Renal Medicine, Royal Prince Alfred Hospital, Sydney, NSW 2050, Australia; julian.singer@sydney.edu.au (J.S.); tracey.ying@sydney.edu.au (T.Y.); huiling.wu@sydney.edu.au (H.W.); 2Central Clinical School, Faculty of Medicine and Health, University of Sydney, Camperdown, NSW 2006, Australia; bree.shi@sydney.edu.au; 3Central and Northern Adelaide Renal and Transplantation Service, Royal Adelaide Hospital, Adelaide, SA 5000, Australia; matthew.tunbridge@sa.gov.au (M.J.T.); tania.salehi@sa.gov.au (T.S.); beatrice.sim@uqconnect.edu.au (B.Z.S.); svjetlana.kireta@adelaide.edu.au (S.K.); julie.johnston@sa.gov.au (J.K.J.); toby.coates@sa.gov.au (P.T.C.); 4Adelaide Medical School, University of Adelaide, Adelaide, SA 5000, Australia; griffith.perkins@adelaide.edu.au (G.B.P.); chengsheng.chai@adelaide.edu.au (C.S.C.); pravin.hissaria@sa.gov.au (P.H.); branka.grubor@adelaide.edu.au (B.G.-B.); 5Immunology Directorate, SA Pathology, Adelaide, SA 5000, Australia; 6Kirby Institute, University of New South Wales, Sydney, NSW 2052, Australia; aakerman@kirby.unsw.edu.au (A.A.); vmilogiannakis@kirby.unsw.edu.au (V.M.); aaggarwal@kirby.unsw.edu.au (A.A.); sturville@kirby.unsw.edu.au (S.T.); 7Department of Immunology and Allergy, Royal Adelaide Hospital, Adelaide, SA 5000, Australia; 8Viral Immunology Group, Basil Hetzel Institute for Translational Health Research, University of Adelaide, Adelaide, SA 5011, Australia

**Keywords:** COVID-19, SARS-CoV-2, kidney transplantation, microbiota, immunisation, prebiotics, dysbiosis, immunity

## Abstract

Kidney transplant recipients are at an increased risk of hospitalisation and death from SARS-CoV-2 infection, and standard two-dose vaccination schedules are typically inadequate to generate protective immunity. Gut dysbiosis, which is common among kidney transplant recipients and known to effect systemic immunity, may be a contributing factor to a lack of vaccine immunogenicity in this at-risk cohort. The gut microbiota modulates vaccine responses, with the production of immunomodulatory short-chain fatty acids by bacteria such as *Bifidobacterium* associated with heightened vaccine responses in both observational and experimental studies. As SCFA-producing populations in the gut microbiota are enhanced by diets rich in non-digestible fibre, dietary supplementation with prebiotic fibre emerges as a potential adjuvant strategy to correct dysbiosis and improve vaccine-induced immunity. In a randomised, double-bind, placebo-controlled trial of 72 kidney transplant recipients, we found dietary supplementation with prebiotic inulin for 4 weeks before and after a third SARS-CoV2 mRNA vaccine to be feasible, tolerable, and safe. Inulin supplementation resulted in an increase in gut *Bifidobacterium*, as determined by 16S RNA sequencing, but did not increase in vitro neutralisation of live SARS-CoV-2 virus at 4 weeks following a third vaccination. Dietary fibre supplementation is a feasible strategy with the potential to enhance vaccine-induced immunity and warrants further investigation.

## 1. Introduction

Infection is a leading cause of death among kidney transplant recipients (KTRs), primarily because of the requirement to take lifelong immunosuppressant medication to prevent transplant rejection [1]. KTRs prove to be particularly susceptible to severe acute respiratory syndrome coronavirus 2 (SARS-CoV-2), exhibiting rates of hospitalisation and mortality that far exceed those observed among the general population [2]. Vaccination is the primary health measure to protect against coronavirus disease 2019 (COVID-19). However, in immunosuppressed KTRs, vaccine-induced immune responses remain suboptimal [3,4,5], with a standard two-dose COVID-19 vaccine schedule providing only limited real-world protection from severe disease [6]. Whilst additional vaccine doses have been shown to increase antibody titres in those with an inadequate response to their primary two-dose vaccine schedule, a substantial minority of KTRs remain unprotected despite booster vaccinations [7,8,9].

Both the quality and durability of the vaccine response is dependent on intrinsic (age, sex, and genetics) and potentially modifiable extrinsic (medications, diet, and behaviour) host factors [10,11]. Whilst reducing or transiently altering immunosuppression in KTRs prior to vaccination may be an effective approach to increase vaccine efficacy [12,13], such a strategy may incur increased risks of rejection [14]. Balancing the need to prevent rejection with the necessity of achieving an effective vaccine response is a complex and delicate task, and novel strategies are needed if KTRs are to be protected from COVID-19-associated morbidity and mortality.

Improving the health of the gut microbiota through dietary prebiotic supplementation may be one such strategy. The commensal microorganisms of the gastrointestinal tract exert far-reaching influences on systemic immunity, playing pivotal roles in both the development and licensing of immune cells and in the maintenance of robust immune responses to encountered antigens, including those introduced through vaccination [11,15,16,17]. Crucially, specific microbiota bacteria, such as short-chain fatty acid (SCFA)-producing *Bifidobacterium*, have been independently linked to enhanced virus-specific antibody responses and heightened reactivity to parental vaccines [18,19], including those directed against SARS-CoV-2 [20].

Gut dysbiosis is common among KTRs, maximal around the time of transplantation but persisting lifelong. A reduced abundance of bacteria capable of digesting non-fermentable dietary fibre to generate short-chain fatty acids (SCFA) is a characteristic finding [21]. Boosting *Bifidobacterium* populations and the production of SCFAs by gut microbes could serve as an effective strategy for enhancing vaccine responses. Prebiotic dietary fibre has been shown to increase the abundance of SCFA-producing bacteria in the gut microbiota [22,23,24] and alter systemic immune responses [25], although the efficacy and tolerance of prebiotic fibre has not been examined in KTRs. Whether the prevalent gut microbiome of KTRs is amenable to dietary interventions and whether correction of dysbiosis can promote the immunogenicity of COVID-19 vaccines are therefore important research questions.

To answer these questions, we performed a prospective multi-centre placebo-controlled randomised trial of dietary inulin in KTRs who had failed to mount an effective serological immune response to a standard two-dose COVID-19 vaccine schedule. In this trial, we evaluated the tolerance, feasibility, and efficacy of inulin supplementation to alter the gut microbiome and enhance the immune response to a third COVID-19 vaccine.

## 2. Materials and Methods

### 2.1. Study Design

RIVASTIM-inulin was a registered, prospective randomised, double-blind, parallel-arm, placebo-controlled, multi-centre trial, and the trial design, the rationale, and pre-specified outcomes have been previously published [26]. Briefly, participants were screened for inclusion according to pre-specified criteria from local transplant recipient databases and during routine clinical review at two tertiary referral hospitals in Australia: (1) The Royal Adelaide Hospital, Adelaide, South Australia, and (2) The Royal Prince Alfred Hospital (RPA), Sydney, New South Wales. The recruitment target was 120 participants. The eligibility criteria required participants to be at least 18 years old with a functional kidney transplant from either a living or deceased donor. Participants must have received two doses of a COVID-19 vaccine and subsequently shown a suboptimal immune response, with SARS-CoV-2 receptor-binding domain antibodies (anti-RBD Ig) falling below the threshold for clinical protection from COVID-19 (<100 units/mL). Individuals were excluded from the study if they had undergone multi-organ transplants (such as kidney-pancreas), were pregnant at the time, had a recorded previous COVID-19 infection, had known allergies or intolerances to dietary fibre, or an underlying gastrointestinal condition. Full eligibility criteria are provided in the clinical trial protocol (Appendix A). All patients provided written informed consent and participants were assigned in a 1:1 ratio to receive either inulin or maltodextrin (control) through computer-generated stratified block randomisation, utilising randomly permuted block sizes of 2, 4, and 6. Stratification was by trial centre and response to a two-dose COVID-19 vaccine schedule (low responder anti-RBD Ig 0.8–99 U/mL; or non-responder, anti-RBD Ig < 0.8 U/mL).

The organisational structure of the study is depicted in Figure 1. Following randomisation, participants were allocated a dietary supplement in the form of a white, soluble, mostly tasteless, and visually indistinguishable powder containing either inulin (active group) or maltodextrin (control group). Participants were instructed to consume 10 g of the supplement dissolved in 200 mL of water daily for one week, after which the dosage was increased to 10 g twice daily. After a 4-week lead-in period, participants received an mRNA COVID-19 vaccine, and their antibody response was measured 4–6 weeks following vaccination. The participants continued taking the dietary supplement until the time of antibody assessment. Baseline characteristics were recorded using trial-specific electronic data capture forms (eDCFs) stored within a secure, web-based data management tool (REDCap) [27]. Blood samples were drawn by clinical research staff for immunological assessment and routine biochemistry at enrolment and at 4–6 weeks following vaccination. Tolerance to the study intervention was assessed by measuring changes in the gastrointestinal symptom rating scale (GSRS) [28,29] at baseline, at week 4, and at the final trial visit. Health-related quality of life information was collected using the EQ-5D index [30]. Adverse events (AEs), including adverse events following immunisation (AEFIs) and adverse events of special interest (AESIs) were evaluated through phone consultations conducted 1 week and 4–6 weeks after vaccination.

### 2.2. Interventions

The study products were specifically prepared and packaged for the trial by Bulk Powders Pty Ltd. (Braeside, VIC, Australia) in identical, sealed, opaque, and numbered 1 kg bags, each accompanied by a 10 g measuring scoop. Adherence to supplementation was assessed by both a questionnaire at the time of the study visit and via weighing of the remaining supplement at study cessation.

Immunisation occurred with a COVID-19 mRNA vaccine, either Pfizer-BioNTech BNT162b2 (30 µg, IM) or Moderna mRNA-1273 (50 µg, IM), determined by local practice and vaccine availability. Participants were instructed to comply with recommended public health measures (e.g., social distancing, hand hygiene, wearing a face mask, and COVID-19 testing and isolation as mandated).

All participants maintained their regular transplant management according to local standards of care and under the guidance of their treating nephrologist. Participants were advised to adhere to their usual diet and refrain from taking any prebiotics or probiotics not included in the study.

### 2.3. Trial Outcomes

The primary endpoint was the percentage of individuals within each experimental group who attained protective serological neutralisation against the live SARS-CoV-2 virus (Wuhan) at 4 to 6 weeks following their third COVID-19 vaccination. Protective immunity was delineated as reaching 20.2% of the average neutralisation level observed in the serum of a standardised group of individuals who had recovered from COVID-19. This benchmark corresponds to a 50% protection from SARS-CoV-2 (Wuhan) infection among healthy individuals [31].

Secondary outcome measures included:The alteration in the median intensity of the SARS-CoV-2 Spike-specific antiviral T cell reaction before and at 4 to 6 weeks post-vaccination, assessed by evaluating the frequency of cells producing IFN-γ when stimulated with Spike protein (Wuhan) derived peptides using the ELISpot method.Inulin tolerance was assessed by changes in the Gastrointestinal Symptom Rating Scale (GSRS) at baseline, week 4, and the concluding trial visit. The occurrence of gastrointestinal symptoms (GIS) was characterised by experiencing at least one symptom or having a GSRS score of ≥2.The percentage of individuals who displayed a serological response at 4–6 weeks following a third COVID-19 vaccination, defined as reaching a cut-off of anti-receptor-binding domain antibody (anti-RBD Ig) ≥ 100 units/mL. This RBD antibody threshold was determined on the evidence of pre-clinical and clinical studies [31,32] and is consistent with the scientific literature in COVID-19 clinical vaccine trials [4,33].Changes in the community composition, the relative abundance, and the operational attributes of the gut microbiome evaluated at 4 weeks following commencement of the intervention, determined through 16S rRNA metagenomic sequencing of stool samples collected from participants.COVID-19 infection following randomisation, determined by either:
Positive SARS-CoV-2 PCR test or rapid antigen test in the setting of symptomatic disease.Detection of SARS-CoV-2 anti-nucleocapsid antibodies at the time of primary outcome assessment.


### 2.4. Sample Size

Whilst observational evidence supports the role of the gut microbiota in modulating the immune response to vaccination, the evidence supporting interventions that target the microbiota in this setting are lacking and estimates of effect size cannot be accurately generated.

We therefor considered the following when defining our sample size: (1) the number of eligible KTRs across the two sites; (2) their current vaccination status; (3) the feasibility of conducting a trial within the contemporary resource setting; (4) the local prevalence of COVID-19; and (5) the recommended sample size requirements for a pilot study [34,35]. A recruitment target of approximately 60–120 participants was considered feasible and sufficient to generate a meaningful estimate of effect size with adequate confidence. With full recruitment of 120 participants, and assuming a 25% virus neutralisation endpoint in the control group [4,13], we would require 54% virus neutralisation in the intervention group to demonstrate superiority using a one-sided hypothesis with 2.5% α-risk and 90% power.

### 2.5. IFN-γ ELISpot

Thawed peripheral blood mononuclear cells (PBMCs) were stimulated for 18 h using four sets of peptide pools that covered the entire sequence of the Spike glycoprotein from the USA-WA1/2020 strain. These peptides, sourced from BEI Resources, NIAID, NIH (Peptide Array, SARS-Related Coronavirus 2 Spike (S) Glycoprotein, NR-52402), consisted of individual peptides ranging from 17 to 13 amino acids in length, with 10 amino acid overlaps.

Using multiscreen-IP HTS plates (Merck Millipore, Darmstadt, Germany) coated with anti-human IFN-γ (clone 2G1, Thermo Fisher Scientific, Waltham, MA, USA), secreted IFN-γ was identified with anti-human IFN-γ biotin (Clone B133.5; Thermo Fisher, Scientific, Waltham, MA, USA) followed by streptavidin–HRP (BD Biosciences, Franklin Lakes, NJ, USA) and AEC substrate (BD Biosciences, Franklin Lakes, NJ, USA). An ELISpot reader (Cellular Technology Ltd., Bonn, Germany) automatically counted developed spots with the number of unstimulated splenocytes (<50) subtracted from the number of spots for the peptide pool-stimulated splenocytes to calculate the figure of specific spot-forming units per 10^6^ cells.

### 2.6. Receptor-Binding Domain Ig

RBD-Ig and nucleocapsid-specific antibodies were quantified using the ‘Elecsys Anti-SARS-CoV-2 S’ and ‘Elecsys antiSARS-CoV-2’ assays on the Cobas system (Roche, Basel, Switzerland), following the manufacturer’s instructions. The measurable range for detecting anti-RBD Ig in this assay is 0.8–250 U/mL. One U/mL is comparable to 1 BAU/mL.

### 2.7. SARS-CoV-2 Live-Virus Neutralisation Assay

In 384-well plates, HEK-ACE2/TMPRSS cells (Clone 24) [36] were seeded at 5 × 10^3^ cells/well with 5% *v*/*v* live cell nuclear stain Hoechst-33342 dye (NucBlue, Invitrogen, Waltham, MA, USA). Two-fold dilutions of patient serum samples were incubated with an equal volume of SARS-CoV-2 virus solution (1.25 × 10^4^ TCID50/mL) at 37 °C for 1 h before adding 40 μL, in duplicate, to the cells (final MOI = 0.05). The key viral variants of concern, Delta (B.1.617.2) and Omicron (B.1.1.529), in addition to ‘wild-type’ control virus (A.2.2) from clade A and with no Spike amino acid mutations (similar to Wuhan ancestral variant) were studied. Plates were incubated for 24 h, wells were imaged by high-content fluorescence microscopy, and cell counts derived by IN Carta Image Analysis Software (V15, Cytiva, Marlborough, MA, USA). Virus neutralisation was calculated with the formula: %N = (D − (1 − Q)) × 100/D, as previously described [36]. Neutralising activity was defined as an average %N > 50%.

### 2.8. Bacterial 16S rRNA Gene Amplicon Sequencing and Bioinformatics

Stool samples were collected by participants in their home using an all-in-one system for collection and microbial DNA stabilisation (OMNIgene GUT OM-200, DNA Genotek, Stittsville, ON, Canada). Returned specimens were aliquoted and stored at −80 °C. DNA from faecal samples were extracted using the QIAamp DNA stool mini kit (QIAGEN), according to the manufacturer’s protocol. The V4 region (515F–806R) of the 16S rRNA gene of microbial DNA was profiled using tagged amplicons on the Illumina MiSeq Platform (2 × 250 bp) at the Ramaciotti Centre for Genomics (University of New South Wales, Sydney, Australia).

The DADA2 pipeline [37] was used to sequence and align reads before being partitioned into amplicon sequence variants (ASVs) using R software (4.3.0) [38]. Low-quality reads were filtered and chimeric sequences removed. Taxonomy was assigned using the Ribosomal Database Project naive Bayesian classifier with species level taxonomy assignment [39]. ASVs that contained less than 0.01% of total reads or less than 0.1% of reads in at least 5% of samples were filtered out. To account for the compositional nature of the dataset, ASVs were scaled with a pseudocount of 1 and data were then centre log-ratio (CLR) transformed [40]. Alpha-diversity metrics were calculated using the R package phyloseq [41] and analysed using mixed models with repeat measures, with timepoint and intervention as fixed effects and subject as a random effect, to compare between group differences. Disparities in microbiome composition among groups (beta diversity) were assessed using a PERMANOVA test based on the Aitchison distance matrix (calculated as the Euclidean distance of CLR-transformed data), after confirming homogeneity of variances with the betadisper function of the R package vegan 2.6.4 [42]. Principal coordinate analysis (PCoA) plots were generated to display the beta diversity. The differential abundance of ASVs was calculated by ALDEX2, with a Benjamini–Hochberg false discovery rate (FDR)-corrected Wilcoxon test *p*-value of <0.05 for significance. The effect size of differentially abundant taxa was reported, as effect size measures have been shown to be more reproducible than *p*-values [43,44]. Using the PiCRUST2 pipeline [45], inferred metagenomic pathways were generated from sequence reads and compared using ALDEx2, as above. Correlations between ASVs and generated metabolic pathways were examined using Spearman’s correlation test, and visualised using the chordDiagram function of the R package circlize [46]. Sequencing data were deposited in the European Nucleotide Archive under accession number PRJEB74460.

### 2.9. Quantification and Statistical Analysis

Participant characteristics are presented as percentages for ordinal variables, and continuous variables are presented as mean ± standard deviation (SD) for normally distributed data and median with interquartile range (IQR) for non-normally distributed data. Baseline characteristics were compared using a *t*-test or a Mann–Whitney U-test, as appropriate. Participants were analysed according to an intention-to-treat principle. To assess the primary outcome, we calculated the risk ratio and its corresponding 95% confidence interval using a generalised linear mixed model. This model employed a binomial distribution and a log-link function, with the study site being accounted for as a random effect. A log-binomial regression model was used to calculate unadjusted and adjusted relative risks, with initial immune response as a fixed effect and study site as a random effect.

Analysis of the change in GSRS scores over the intervention period was performed using mixed models with repeated measures for both the global GSRS score and for each GSRS domain.

All statistical analyses were conducted using R version 4.3.0 [38], while figures were generated utilising the ggplot2 [47] and ComplexHeatmap [48] packages. Two-sided statistical tests were employed, and a *p*-value or false discovery rate-adjusted *q*-value < 0.05 was regarded as statistically significant.

## 3. Results

### 3.1. Study Design

To investigate the role of prebiotic inulin supplementation in enhancing the immune response to a third COVID-19 vaccination in kidney transplant recipients (KTRs), we conducted a prospective randomised, double-blind, parallel-arm, placebo-controlled, multi-centre trial. The trial outline is demonstrated in Figure 1. Between 11 November 2021 and 18 May 2022, we enrolled 94 double-vaccinated KTRs, among whom 22 (23%) displayed adequate protective immunity to a primary vaccination course (anti-RBD Ig ≥ 100 U/mL) and were screened out. The remaining 72 KTRs proceeded to randomisation and were allocated to receive 10 g/daily (escalating to 20 g/daily after one week) of either dietary inulin (*n* = 37) or a matched placebo (*n* = 35). In each study arm, 2 participants withdrew consent and 1 patient in the inulin arm and 2 in the placebo arm developed symptomatic COVID-19 and were excluded from the analysis (Figure 2). The baseline characteristics of the study population are reported in Table 1. Participants were predominantly male (51/72, 71%), with mean age of 58 ± 11 years. The median time post-transplant was 7.9 [2.5–13.9] years and the mean eGFR was 56.1 ± 25.9 mL/min. The primary vaccination schedule was with BNT162b2 (Pfizer-BioNTech) in 42/72 (58%) and ChAdOx1-S (AstraZeneca) in 30/72 (42%). In the majority of patients, immunosuppression was with mycophenolate and prednisone in combination with either a calcineurin inhibitor (43%) or mTOR inhibitor (15%) (Appendix A). A total of 65 participants received a third COVID-19 vaccination at a median of 31 [28–37] days following the intervention, among whom 98% received BNT162b2 (Pfizer-BioNTech).

### 3.2. Inulin Supplementation in Kidney Transplant Recipients Is Feasible and Well Tolerated

Patients allocated to supplementation with dietary inulin or placebo displayed similar adherence (85% vs. 72%, *p* = 0.24) and tolerance (97% vs. 94%, *p* = 0.94), respectively, to the prescribed intervention (Table 2). At baseline, gastrointestinal symptoms were common, with 69% and 73% of participants assigned to the inulin or placebo group, respectively, reporting at least one gastrointestinal symptom. The most frequently reported symptoms were indigestion (59% vs. 50%, *p* = 0.52) and diarrhoea (41% vs. 38%, *p* = 0.82, Appendix A). Following 8 weeks of intervention, there was no statistically significant difference in the estimated global GSRS score between treatment groups (*p* = 0.07). However, inulin supplementation was associated with an increase in indigestion score (estimated difference, 2.34 (CI: 0.39–4.48, *p* = 0.02). Inulin supplementation was not associated with a change in patient-reported overall health status, assessed using the EQ-5D-5L index (inulin, 0.91 ± 0.13 vs. placebo, 0.95 ± 0.08; *p* = 0.24. Appendix A).

There was no significant difference in the frequency of adverse events between the inulin and placebo groups (41% vs. 29%, *p* = 0.31, Appendix A). There were no episodes of acute allograft rejection, and no significant change in estimated GFR or albuminuria occurred in either group (Appendix A).

### 3.3. Inulin Supplementation Did Not Alter the Humoral Immune Response to a Third COVID-19 Vac-Cination

Following a third vaccination, 21 of 32 patients (66%) in the inulin group had protective neutralisation titres against SARS-CoV2 (ancestral strain A2.2.2), as compared with 18 of 29 patients (62%) in the placebo group (risk ratio [RR], 1.06; 95% confidence interval [CI], 0.72 to 1.54; *p* = 0.77, Figure 3A). Similarly, the capacity of sera to neutralise Omicron BA.5 strains was comparable in both groups, although less than that seen against the A2.2.2 strain (inulin, 47% vs. placebo, 52%, RR, 0.87; 95% CI 0.53 to 1.45, *p* = 0.59, Figure 3B). Consistent with previous reports, our cohort showed a strong correlation between neutralising antibodies by live virus neutralisation and anti-Spike RBD Ig (Spearman’s r = 0.47, *p* < 0.003, Appendix A). There was no difference in anti-RBD seroconversion between the inulin and placebo groups (RR 1.01, 95% CI: 0.61–1.70, *p* = 0.96, Figure 3C), which persisted following adjustment for baseline anti-RBD levels (Appendix A). However, in a multivariate model, KTRs with a low response (SARS-CoV-2 anti-RBD Ig > 0.8 but < 100 U/mL) following two vaccine doses were significantly more likely to develop protective neutralisation following a third vaccine dose than KTRs with no detectable humoral response to a primary vaccination course (RR 3.20, 95% CI: 1.14–9.45, *p* = 0.03, Figure 3D). T-cell responses were determined by IFN-γ ELISpot both preceding and following booster vaccination (Figure 3E), with the median change in T-cell response following a third vaccination remaining unaltered between groups (*p* = 0.74, Figure 3F).

### 3.4. Inulin Did Not Significantly Alter the Alpha-Diversity of the Gut Microbiota after 4 Weeks of Dietary Supplementation

Inulin supplementation has been shown to alter the gut microbiome by promoting the select growth of bacteria capable of utilising non-digestible fibre through fermentation. To determine whether inulin supplementation altered the gut microbiome, we collected 78 faecal samples from 39 participants, including 18 patients in the inulin group and 21 patients in the control group; a baseline sample was collected prior to randomisation, and a second sample was collected following 4 weeks of the intervention. The gut microbiota was characterised through sequencing of the hypervariable V4 region of the bacterial 16S rRNA gene. Following quality and prevalence filtering, a total of 1,905,711 high-quality sequence reads were included (24,432 ± 7125 sequences per sample), generating 2025 unique amplicon sequence variants (ASVs).

At baseline, the microbiota of KTRs demonstrated marked heterogeneity, although Firmicutes was the dominant phyla in 33 (85%) of the participants, with a mean relative abundance of 60.8 ± 15.2% (Figure 4A). Compared to the placebo, inulin supplementation resulted in a decrease in gut microbiota richness, diversity, and evenness, although these results did not reach statistically significance (Figure 4B, Appendix A).

### 3.5. Prebiotic Inulin Did Not Significantly Alter the Community Composition of the Gut Microbiome

The beta diversity of the gut microbiota was assessed by principal coordinate analysis (PCA) of Aitchison distances between individual samples at baseline and following 4 weeks of the treatment intervention. Neither supplementation with inulin or placebo significantly shifted the community composition (PERMANOVA, 0.99, Figure 4C,D) and, at the time of vaccination, the community composition remained similar between the inulin and placebo groups (Figure 4E, *p* = 0.32).

We then investigated whether the community composition at the time of a third COVID-19 vaccine was associated with the development of effective SARS-CoV-2 viral neutralisation; however, the beta diversity between vaccine responders and non-responders remained similar when assessed using both phylogenetic and non-phylogenetic measures (Appendix A).

### 3.6. Inulin Supplementation Increases Bacterial Genus Abundance and Promotes SCFA-Producing Bifidobacterium Species

As prebiotic inulin supplementation has been shown to promote the growth of key microbial species, we examined the relative abundance of the microbiota in inulin- and placebo-supplemented KTRs at the time of a third SARS-CoV-2 vaccination. The gut microbiota composition at the family level is shown in Figure 4F.

At the genus level, inulin supplementation resulted in a statistically significant increase in short-chain fatty acid (SCFA)-producing bacteria in the gut microbiota, with *Bifidobacterium*, a prototypical SCFA-producing bacterium, showing a 2.3-fold increase in relative abundance (Figure 4G, *p* < 0.001) compared to the placebo group. Other SCFA-producing bacteria, such as Anaerostipes, a butyrate-producing bacterium from the family Lachnospiraceae, and Caproiciproducens, an anaerobic bacterium that produces acetate, butyrate, and caproate [49], were similarly more prevalent in inulin-supplemented KTRs. However, using the robust ALDEx2 analysis, only *Bifidobacterium* remained significantly differentially abundant (Figure 4H and Appendix A). As the abundance of *Bifidobacterium* has been linked to an enhanced immune response to coronavirus vaccination in previous reports [20], we subsequently investigated whether there was an association between the relative abundance of *Bifidobacterium* and the response to a third SARS-CoV-2 vaccination. Spearman’s correlation analysis revealed the abundance of *Bifidobacterium* to be significantly correlated with inulin supplementation (rs = 0.6, *p* = 0.028) but not with the development of protective neutralisation of live Wuhan strain virus or Spike-specific T-cell response (Appendix A). Together, these results confirmed dietary inulin as an effective prebiotic therapy to augment SCFA-producing bacteria in the gut microbiota of KTRs.

### 3.7. Inulin Supplementation Leads to an Alteration in the Metagenomic Function of the Gut Microbiota

Dietary inulin has been shown to alter host immunity through the production of microbiota-derived immunomodulatory metabolites, such as SCFA, a proposed mechanism by which the microbiota may improve vaccine immunogenicity [20]. We therefore examined the predictive metagenomic function of the gut microbiota in response to inulin supplementation using PiCRUST2 to identify Kyoto Encyclopaedia of Genes and Genomes (KEGG) orthologues (KO) at the gene level. The various KEGG orthologues were mapped into KEGG pathways, and the differential abundance of enriched pathways was compared using ALDEx2. We found inulin supplementation to significantly increase metabolic pathways involved in the microbial production of SCFAs (Figure 5A) and used Spearman’s correlation analysis to reveal the key bacteria in the gut microbiota associated with the observed enhanced metabolic pathways (Figure 5B).

## 4. Discussion

In this placebo-controlled and randomised trial, we successfully established the tolerability and feasibility of dietary inulin in a cohort with a high prevalence of gastrointestinal symptoms. Inulin proved effective in promoting the growth of short-chain fatty acid (SCFA)-producing gut bacteria, notably *Bifidobacterium*; however, there was no difference in the humoral or cellular immune responses between treatment groups following a third COVID-19 vaccine.

The necessity for kidney transplant recipients (KTRs) to undergo lifelong immunosuppression presents a host of formidable challenges, including an increased susceptibility to life-threatening infection and a variable and often inadequate response to vaccination [1,50,51,52]. Whilst the inadequacy of vaccine responses has long been acknowledged and tolerated, the emergence of the COVID-19 pandemic has emphasised the severity of these issues, with KTRs experiencing a 24% mortality rate in the early phase of the pandemic [53] and limited real-world protection by a standard vaccination course [6]. Several strategies have been employed in an attempt to generate protective immunity in KTRs, with priority access to vaccination [54,55], successive rounds of booster vaccination [4,56,57,58,59,60], ring vaccination of close contacts [61,62], and prophylactic monoclonal antibody therapy [63,64] all employed with varying degrees of efficacy. Booster vaccination, for example, increases the proportion of KTRs with protective levels of neutralising antibodies, with a third dose generating functional humoral immune responses in up to 60% of KTRs [4,8,9]. However, an important minority remain seronegative despite a fourth [57,58,65] or fifth COVID-19 vaccine [66]. Furthermore, the durability of both humoral and cellular immunity wanes significantly with time, particularly in KTRs [67,68].

Given the suboptimal efficacy of COVID-19 vaccination and the current adjuvant therapies and strategies available, there is an unmet need for a novel, tolerable, and feasible therapy to boost vaccine responses in at-risk cohorts. Such a therapy has the potential to reduce the hospitalisation, mortality, and graft injury associated with COVID-19 infection in KTRs.

Kidney transplantation results in a significant and sustained alteration of the gut microbiome, with this dysbiosis characterised by a loss of microbial diversity and a decrease in important metabolic pathways, including SCFA production [21]. Similar dysbiosis, when observed in the elderly or disparate populations between low-income and middle-income countries, is associated with functional deficits in immune responses, including toward vaccination [11,69]. Recently, two prospective studies identified distinctions in the composition of the microbiome between individuals with high and low humoral responsiveness to BNT162b2 (Pfizer BioNTech) vaccination. In their study, Ng et al. identified a correlation between the abundance of *Eubacterium rectale*, Roseburia faces, *Bacteroides thetaiotaomicron*, and *Bacteroides* spp. OM05-12 with a heightened BNT162b2 vaccine response in a population of largely healthy volunteers [20]. By contrast, participants with a low immune response to BNT162b2 had a persistently a low level of Actinobacteria, particularly *Bifidobacterium* spp. Similarly, Healey et al. observed populations of *Prevotella*, *Haemophilus*, *Veillonella*, and *Ruminococcus gnavus* to be abundant in high responders to the BNT162b2 vaccine within a cohort of healthcare workers [70]. Whilst both of these studies support the role of the microbiome in modulating the antibody response to SARS-CoV-2 vaccination, the specific bacterial taxa associated with a heightened vaccine response were unique in each cohort. Variances in microbial populations can be attributed to differences in geography and diet among cohorts, yet together these findings suggest that functional, rather than taxonomic changes, are the drivers of an altered humoral response to vaccination. In this regard, Healey et al. [70] examined potential links between the functional capacity of the microbiome with vaccine responses, observing that microbe-derived branched chain fatty acids (BCFAs) formed via microbial protein fermentation were increased in low vaccine responders. Supporting this finding, they noted the abundance of prominent BCFA producers, *Megasphaera* spp., to be similarly negatively associated with vaccine responses. By contrast, intake of dietary fibre, a microbial substrate for SCFA generation, was associated with greater vaccine-induced IgG binding avidity, supporting the role of microbial metabolites in modulating systemic immune responses.

T-cell responses to vaccination, although less well explored, are also crucial for effective protection. Supporting evidence is provided from patients with impaired antibody responses due to B-cell depletion, with greater protection from COVID-19 afforded to those with elevated CD8+ T-cell counts [71]. In the present study, inulin supplementation did not result in a significant increase in the median T-cell response to a booster COVID-19 vaccination, as assessed by IFN-γ ELISpot. However, a significant within group difference was observed in the inulin cohort, but not the placebo cohort, following a third vaccination, suggesting a potential role for microbiome manipulation in augmenting the cellular response to vaccination. This hypothesis was supported by recent animal work comparing the immune responses in germ-free (GF) and specific pathogen-free (SPF) mice subjected to mRNA-LNP vaccination [72]. Here, Johnson et al. observed GF mice to exhibit reduced CD8+ T-cell and innate immune responses to SARS-CoV-2 mRNA vaccination when compared to SPF mice [72]. Interestingly, whilst GF mice showed only a small increase in CD11c expression on monocytes following mRNA-LNP vaccination, SPF mice exhibited a dramatic upregulation in CD11c+ monocytes following vaccination. Crucially, vaccine-induced monocyte activation in SPF mice could be abrogated by antibiotic depletion of the microbiome prior to vaccination. Furthermore, the authors identified the type I IFN signalling pathway as the potential driver of the difference in innate myeloid response seen between GF and SPF following mRNA-LNP immunisation. By contrast, a second study in mice did not find a difference in response in antibiotic-treated or germ-free mice following BNT162b2 vaccination [73]. This second study, whilst illustrating that the microbiota is not essential for vaccine-induced immunity, does not preclude the possibility that specific bacteria in the gut microbiota may influence and enhance the vaccine response. Taken together, the data support a role for the gut microbiome in modulating the immune response to mRNA vaccination, with reduced dendritic cell/macrophage activation and an impaired IFN-I response a potential underlying mechanism.

To our knowledge, this trial provides the first evidence for the capacity of prebiotics to alter the relative abundance of key bacteria in the gut of KTRs, particularly *Bifidobacterium* spp., with further trials ongoing [74]. Inulin produced a measurable biological effect, a crucial first step for dietary prebiotics to alter systemic immunity. Given the high prevalence of gastrointestinal symptoms in KTRs [75], demonstrating the tolerability of inulin supplementation at a dose sufficient to alter the microbiome constitutes a necessary step on the pathway toward the development of a targeted microbiome therapy.

Our trial is not without limitations. Firstly, the trial failed to reach its recruitment target and was hence underpowered to detect a difference in the primary endpoint. Recruitment began at the height of the COVI19 pandemic and was hampered by the emergence of the Omicron variant. The resulting increase in both hospitalisations and public health restrictions led to a rapid upsurge in community access to vaccination, with many KTRs accessing a third vaccine outside of their transplant centre. For many others, delaying access to a third booster vaccine became ethically unacceptable. Secondly, several studies have demonstrated a statistically significant reduction in alpha-diversity and a shift in microbiome composition following the initial BNT162b2 vaccination [20,70]. Whilst we employed a targeted therapy following a standard two-dose vaccine schedule, manipulating the microbiome prior to an initial vaccination may be more effective in promoting vaccine immunogenicity. Thirdly, using maltodextrin as an inert placebo presents certain issues. Maltodextrin is often chosen as a placebo in interventional microbiota trials due to its neutral taste, easy digestibility, and similar physical properties to the investigational product, as seen in this trial. While it was traditionally believed to have minimal impact on gut health and the colonic microbiota due to its rapid absorption in the small intestine, several studies suggest otherwise. In animal research, maltodextrin has been shown to contribute to intestinal inflammation by disrupting the gut mucous gel layer. Clinical studies in humans also indicate that maltodextrin can affect the gut microbiota’s composition, although there is significant variability in the specific bacteria affected and the direction of these changes across different studies. Despite these concerns, no suitable alternative to maltodextrin has been identified or commonly used in microbiota trials. Omitting a control arm from the trial was considered, but it was determined that this would significantly undermine the study’s scientific validity. Lastly, as the factors that influence vaccine responses are manifold, with baseline microbiome composition, immunosuppressive medication regimen, nutritional status, host genetics, age-related immune senescence, and vaccine-associated characteristics all contributing to interpersonal variation [10,16], microbiome manipulation alone may not lead to substantial improvement in vaccine response for all vaccines or for all individuals. Manipulation of the microbiome to enhance vaccine responses may produce only modest effects, and combining microbiome modulation with traditional adjuvants or other immunostimulatory agents could lead to more robust and predictable results. In KTRs, for instance, adjuvants targeting the microbiome may complement additional strategies, including immunosuppression minimisation or alteration, to augment vaccine responses. Crucially, microbiome fortification may allow for a minimal reduction in immunosuppression, with the potential to offer some protection from allograft rejection [76].

Manipulating the microbiome with prebiotic therapy may indeed hold promise for improving vaccine efficacy, but further enquiry, including larger clinical trials, are required to better understand its potential benefits and limitations.

## Figures and Tables

**Figure 1 vaccines-12-00608-f001:**
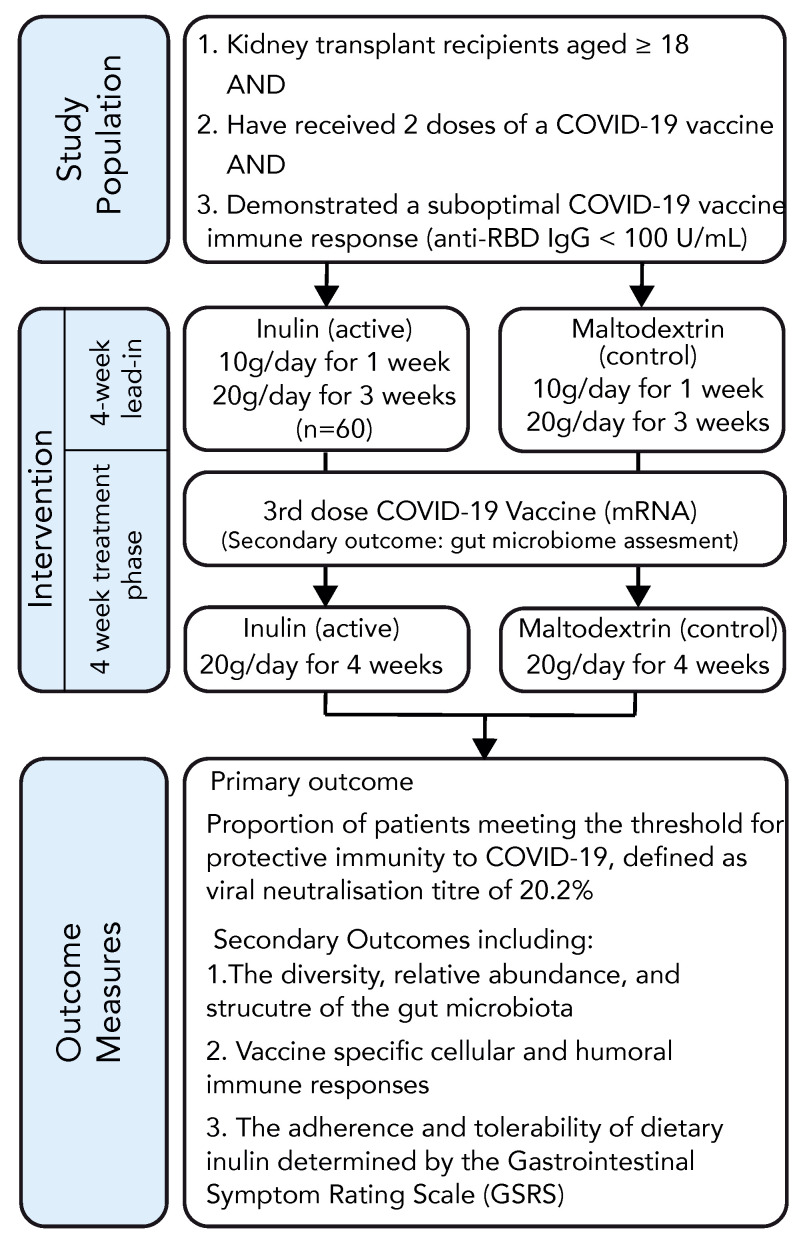
RIVASTIM-Inulin trial design.

**Figure 2 vaccines-12-00608-f002:**
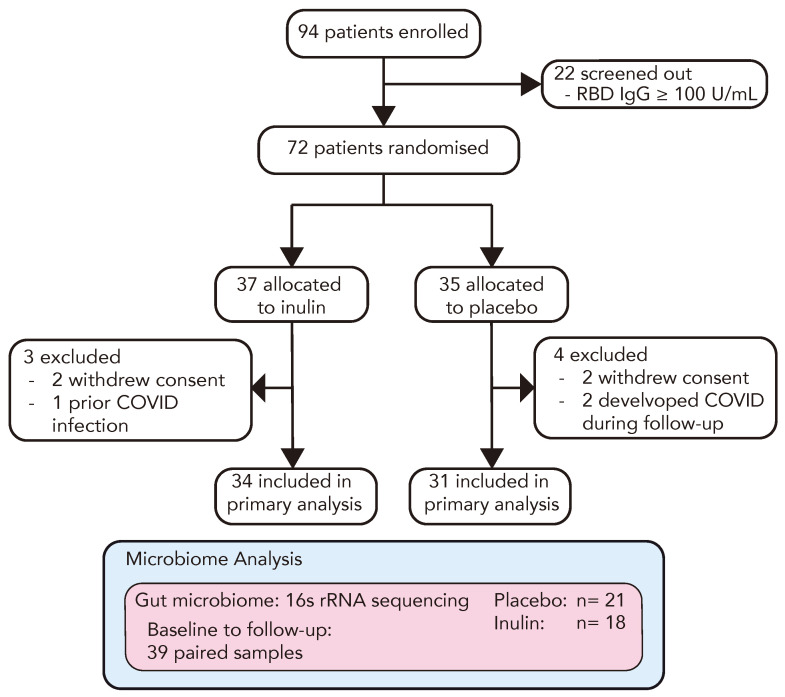
CONSORT flow diagram of the RIAVSTIM-inulin trial. KTRs were screened for inadequate immunity following two doses of SARS-CoV-2 vaccination and randomly allocated to receive dietary supplementation with inulin (*n* = 37) or placebo (*n* = 35). Following 4 weeks of supplementation, participants received a third SARS-CoV-2 vaccine and the subsequent immune response was assessed in 65 participants at 4–6 weeks post-vaccination. A total of 39 participants in the RIVSTIM-inulin trial provided faecal samples for microbiota analysis at baseline and after 4 weeks of dietary supplementation with either inulin or placebo.

**Figure 3 vaccines-12-00608-f003:**
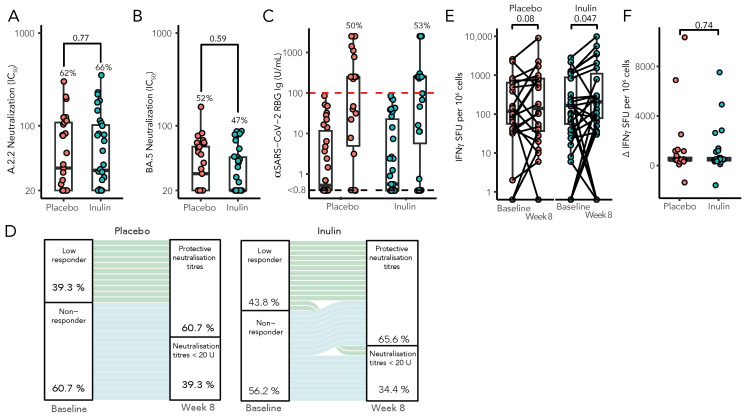
Immunological results of the RIVASTIM-Inulin trial. KTRs with an unsatisfactory response to a primary two-dose COVID-19 vaccination schedule (anti-RBD Ig < 100 U/mL) were randomised to supplementation with dietary inulin or placebo 4 weeks prior to a third dose of mRNA COVID-19 vaccine. At 4 weeks following vaccination, serum neutralisation against live SARS-CoV-2 was assessed for both the ancestral A.2.2 strain (**A**) and the BA.5 strain (**B**), with results expressed as log IC50 values. (**C**) A comparison was made between pre- and post-vaccination anti-RBD Ig titres (U/mL) in both the inulin and placebo groups, with no significant distinction observed in the proportion of patients achieving the predefined target threshold of 100 U/mL between the two groups (*p* = 0.96). (**D**) Compared to non-responders, patients with an initial low response to a primary vaccination course were significantly more likely to develop protective neutralisation following a third vaccination (RR 2.71, 95% CI: 1.37–5.37, *p* = 0.004). (**E**) Inulin supplementation increased Spike-specific T-cell responses as measured by IFN-γ ELISpot (SFU/106 cells), following a third vaccination (Wilcoxon signed-rank test); however, there was no significant difference in the median change in Spike-specific T-cell response between treatment groups by quantile regression (**F**) (*n* = 56, *p* = 0.74).

**Figure 4 vaccines-12-00608-f004:**
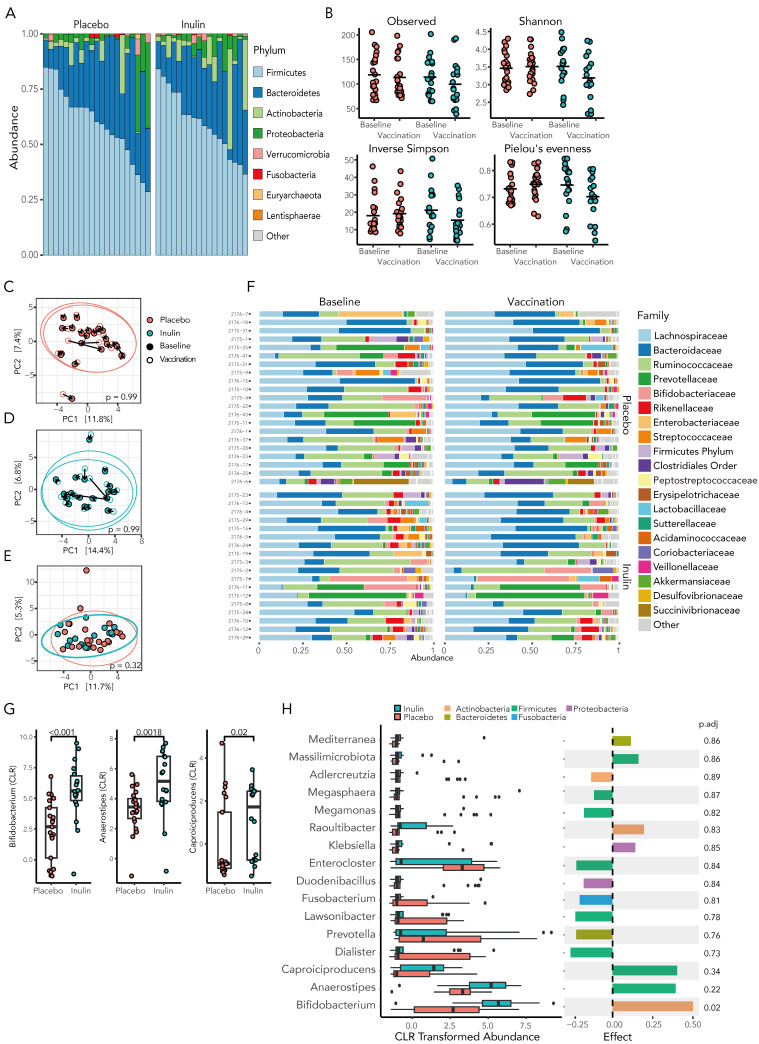
Inulin supplementation in KTRs results in select changes to the gut microbiota. (**A**) The microbiota of KTRs at baseline demonstrates marked heterogeneity although is dominated by the phylum Firmicutes in the majority of KTRs. (**B**) The alpha-diversity of the gut microbiota, as assessed by richness, Shannon diversity index, Inverse Simpson, and evenness, did not differ significantly between groups. Beta diversity, as assessed by principal coordinate analysis (PCA) of Aitchison distances, in individuals at baseline and after 4 weeks of treatment did not differ significantly following placebo (**C**) or inulin (**D**) supplementation. (**E**) At the time of a third SARS-CoV-2 vaccination, there was no significant difference in the microbial community composition between treatment groups. (**F**) The relative abundance of the gut microbiota at the family level is demonstrated at baseline and following 4 weeks of dietary supplementation. (**G**) Inulin supplementation increased the relative abundance of key SCFA-producing bacteria at the genus level by Mann–Whitney U-test and by ALDEx2 (**H**) following correction for multiple comparisons.

**Figure 5 vaccines-12-00608-f005:**
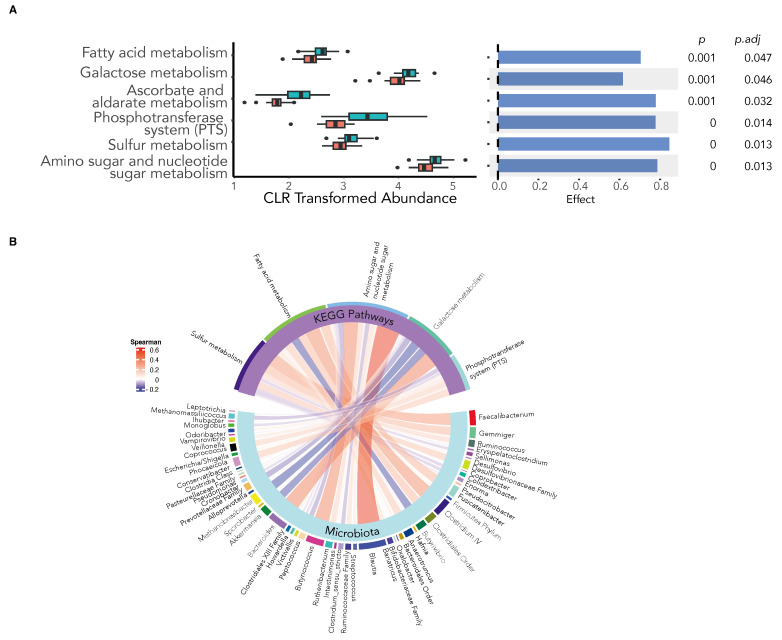
Dietary inulin supplementation alters the metagenomic function of the gut microbiota. KEGG orthologue (KO) genes were predicted using PiCRUST2 and mapped to KEGG pathways. (**A**) The differential abundance of KEGG pathways between the inulin and placebo groups show an increase in the expression of pathways involved in microbial SCFA production following inulin supplementation (ALDEx2 Benjamini–Hochberg-corrected expected *p*-value < 0.05). (**B**) Chord diagram depicting microbial associations with the differentially abundant KEGG pathways. (Spearman’s correlation, Benjamini–Hochberg-corrected expected *p*-value < 0.05).

**Table 1 vaccines-12-00608-t001:** Demographic and clinical characteristics of the patients at baseline.

	Inulin	Placebo	*p*-Value
	*n* = 37	*n* = 35	
Age (mean ± sd)	58 ± 11	60 ± 12	0.31
Sex			0.58
Female	11 (30)	10 (29)	
Male	26 (70)	25 (71)	
BMI at enrolment			0.76
Normal (18 ≤ 25)	13 (35)	11 (31)	
Overweight (25 ≤ 30)	15 (41)	15 (43)	
Obese (>30)	9 (24)	8 (23)	
Missing	0 (0)	1 (3)	
Site			0.68
Royal Adelaide Hospital	26 (70)	23 (66)	
Royal Prince Alfred Hospital	11 (30)	12 (34)	
Self-reported ethnicity			0.54
Caucasian	29 (78)	28 (80)	
Asian	6 (16)	5 (14)	
Other	2 (6)	2 (6)	
Time since most recent Ktx (years)			0.44
0–5	14 (38)	13 (37)	
5–10	6 (16)	8 (23)	
>10	17 (46)	12 (34)	
Missing	0 (0)	2 (6)	
Immunosuppression			0.66
Mycophenolate	24 (65)	27 (77)	
CNI	27 (73)	28 (80)	
mTORi	13 (35)	8 (22)	
Prednisone	33 (89)	34 (92)	
Diabetes Mellitus	13 (35)	9 (26)	0.39
eGFR (mL/min, mean ± sd)	57 ± 31	55 ± 19	0.84
eGFR			0.15
15–29 mL/min	6 (16)	2 (6)	
30–59 mL/min	16 (43)	21 (60)	
60–89 mL/min	10 (27)	11 (31)	
>90 mL/min	5 (14)	1 (3)	
Urine Albumin: Cr ratio (mg/mol, mean ± sd)	59 (116)	21 (58)	0.098
Number of previous kidney transplants			0.15
First graft	25 (68)	28 (80)	
Second or greater	12 (32)	5 (14)	
Missing	0 (0)	2 (6)	
Transplant type			0.44
Deceased	25 (68)	21 (60)	
Living	12 (32)	14 (40)	
Primary Renal Disease			0.96
Glomerulonephritis	13 (35)	10 (29)	
Diabetes mellitus	3 (8)	3 (9)	
Polycystic kidney disease	4 (11)	3 (9)	
Hypertension/renovascular disease	2 (5)	2 (6)	
Vaccination Dose 1			0.26
BNT162b2 (Pfizer Comirnaty)	24 (65)	18 (51)	
ChAdOx1-S (AstraZeneca)	13 (35)	17 (49)	
Baseline Anti-RBD Ig			0.88
non-responder (<0.8 units/mL)	20 (54)	21 (60)	
low-responder (≥0.8 units/mL)	17 (46)	14 (40)	

**Table 2 vaccines-12-00608-t002:** Adherence and tolerance to dietary supplementation during the RIVASTIM-inulin trial.

Outcome	Overall (*n* = 65)	Inulin (*n* = 34)	Placebo (*n* = 31)	*p*-Value
Adherence (80% of prescribed therapy)	59/65 (91%)	29/34 (85%)	30/31 (97%)	0.24
Tolerance (% who continued therapy)	62/65 (95%)	33/34 (97%)	29/31 (94%)	0.94

## Data Availability

16S rRNA gene amplicon sequencing data have been deposited at European Nucleotide Archive and are publicly available as of the 3 April 2024, under accession number PRJEB74460. Available at https://www.ebi.ac.uk/ena/. This paper does not report original code. Any additional information required to reanalyse the data reported in this paper is available from the lead contact upon reasonable request, pending ethical approval.

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
