# Peer review of "Dietary Inulin to Improve SARS-CoV-2 Vaccine Response in Kidney Transplant Recipients: The RIVASTIM-Inulin Randomised Controlled Trial"

_vaccines, 2024, doi:10.3390/vaccines12060608_

Round 1

Reviewer 1 Report

Comments and Suggestions for Authors

This paper clearly introduced the meaningful effect of dietary supplementation with inulin on immune responses of kidney transplant recipients to SARS-CoV2 mRNA vaccine in a well-designed trial. They found that inulin supplementation brought an increase in gut Bifidobacterium and improved vaccine-induced immunity to some extents, and the results suggested that the dietary supplementation with pre-biotic inulin before and after a third SARS-CoV2 mRNA vaccine could be developed into a promising practicable strategy to enhance the specific immunity. This trial firstly provides the proof for the prebiotics to increase the relative abundance of critical bacteria in the gut of KTRs, implying the potential for promoting systemic immunity and relieving the gastrointestinal symptoms in KTRs. Their findings would facilitate the development of a targeted microbiome therapy for the safe and feasible control of the infectious diseases in KTRs. Therefore, this manuscript is acceptable for publication in vaccines.

Author Response

We thank the reviewer for their critique and analysis of our manuscript.

Reviewer 2 Report

Comments and Suggestions for Authors

This manuscript delineates the effect of inulin on improving the immune response to SARS-CoV-2 vaccination in kidney transplant recipients. Overall manuscript was well-written, sound, and organised. However, the interpretation of immunologic assessment needs to be addressed.

Major concerns.

1. Receptor binding domain IgG: wrong target statement.
The Elecsys Anti-SARS-CoV-2 S determine all isotypes of the immunoglobulin, including IgG.  Suggest revising this issue and using "Ig" throughout the manuscript.

References:

- (see "Test principle pg.2) https://www.fda.gov/media/144037/download

2. Control group: Why did the control group use maltodextrin as a placebo?

Various literature delineates recognised maltodextrin as a prebiotic too.

Suggests clarifying why this study uses maltodextrin as a placebo and the effect of this substance on the gutmicrobiota. Was it unlike inulin?

3. Table 1: Ig response criteria.
The Elecsys Anti-SARS-CoV-2 has a detection limit of <0.4 U/mL and a cutoff of 0.8 U/mL.

Why did you consider <0.4 U/mL as a non-responder?
The instrument should consider the outcome between 0.40-0.79 U/mL as a seronegative.

4. Were participants intake antibiotics or probiotics (e.g. yoghurt, kimchi, kombucha...) during follow-up? Or do some participants receive faecal transplantation?
This point may alter the outcome.

Comments.

1. Keywords: Suggest adding "immunity" to the keywords to gain searchability.

2. Table 1: The immunity is age-dependent. The elderly group seems to have a poor immune response to vaccination compared with younger groups.

Suggests subgrouping of age groups to make data more informative.
You may be divided into <60 and ≥60 or whatever, depending on your data.

Typos.

1. "AstraZeneca" (no spacing between Astra and Zeneca).

2. Line 252: q-value.

Author Response

This manuscript delineates the effect of inulin on improving the immune response to SARS-CoV-2 vaccination in kidney transplant recipients. Overall manuscript was well-written, sound, and organised. However, the interpretation of immunologic assessment needs to be addressed.

We thank the reviewer for their critique and analysis of our manuscript. The insightful feedback and required revisions have, in our opinion, strengthened the manuscript.

Major concerns.

  1. Receptor binding domain IgG: wrong target statement.

The Elecsys Anti-SARS-CoV-2 S determine all isotypes of the immunoglobulin, including IgG.  Suggest revising this issue and using "Ig" throughout the manuscript.

We thank the reviewer for identifying this error in our methods. We have corrected all references to IgG to Ig in relation to antibodies detected using the Elecsys Anti-SARS-CoV-2 S method.

References:

- (see "Test principle pg.2) https://www.fda.gov/media/144037/download

  1. Control group: Why did the control group use maltodextrin as a placebo?

Various literature delineates recognised maltodextrin as a prebiotic too.

Suggests clarifying why this study uses maltodextrin as a placebo and the effect of this substance on the gutmicrobiota. Was it unlike inulin?

Identifying an inert substance to use as a placebo for investigational microbiota trials remains a challenge, and one that was given much thought in our trial design. Maltodextrin is often used as a placebo due to its neutral taste, easy digestibility, and minimal impact on metabolic parameters at doses typically used in clinical trials. In our trial design, maltodextrin (placebo) was visually indistinguishable from the investigational product (inulin), and shared similar physical characteristics (solubility, taste), ensuring adequate masking of participants and investigators.

We acknowledge that some trials (largely in animal models) have identified maltodextrin as having a role in promoting intestinal inflammation through alterations in the mucous gel layer.1 However, major alterations in the composition of the microbiota have not been consistently observed2, and often not at the doses used in clinical trials. In human use, a recent metanalysis has suggested maltodextrin to impact human physiology and gut microbiota, however, the individual studies suggest considerable variability in the microbiota impacted, and the direction of the registered effect.3 Lastly, we believe the use of a placebo control arm was required to enhance study credibility, minimise bias, standardize care between groups, reduce protocol violations, and meet essential ethical considerations, and no other suitable alternative to maltodextrin is available.

  1. Laudisi F., Di Fusco D., Dinallo V., Stolfi C., Di Grazia A., Marafini I., Colantoni A., Ortenzi A., Alteri C., Guerrieri F., Mavilio M., Ceccherini-Silberstein F., Federici M., MacDonald T.T., Monteleone I., Monteleone G. The food additive maltodextrin promotes endoplasmic reticulum stress–driven mucus depletion and exacerbates intestinal inflammation. Cell Mol Gastroenterol Hepatol. 2019;7:457–473
  2. Arnold AR, Chassaing B. Maltodextrin, Modern Stressor of the Intestinal Environment. Cell Mol Gastroenterol Hepatol. 2019;7(2):475-476. doi: 10.1016/j.jcmgh.2018.09.014. Epub 2018 Oct 17. PMID: 30827413; PMCID: PMC6409436.
  3. Almutairi R, Basson AR, Wearsh P, Cominelli F, Rodriguez-Palacios A. Validity of food additive maltodextrin as placebo and effects on human gut physiology: systematic review of placebo-controlled clinical trials. Eur J Nutr. 2022 Sep;61(6):2853-2871. doi: 10.1007/s00394-022-02802-5. Epub 2022 Mar 1. Erratum in: Eur J Nutr. 2023 Aug;62(5):2345. PMID: 35230477; PMCID: PMC9835112.
  4. Table 1: Ig response criteria.

The Elecsys Anti-SARS-CoV-2 has a detection limit of <0.4 U/mL and a cutoff of 0.8 U/mL.

Why did you consider <0.4 U/mL as a non-responder?

The instrument should consider the outcome between 0.40-0.79 U/mL as a seronegative.

We thank the reviewer for identifying this error. We have changed the threshold in the manuscript from 0.4 to 0.8, so that all values < 0.8 are identified as seronegative, and adjusted the figures and tables accordingly, where required.

  1. Were participants intake antibiotics or probiotics (e.g. yoghurt, kimchi, kombucha...) during follow-up? Or do some participants receive faecal transplantation?

This point may alter the outcome.

No patients received faecal transplants during the study period. The majority of patients were taking Bactrim as PJP prophylaxis prior to and throughout the study. Patients were instructed not to alter from their habitual diet and abstain from dietary supplementation with non-study pre- or pro-biotics. This point is listed in the methods (line 139)

Comments.

  1. Keywords: Suggest adding "immunity" to the keywords to gain searchability.

Thank you for this suggestion, this has been actioned.

  1. Table 1: The immunity is age-dependent. The elderly group seems to have a poor immune response to vaccination compared with younger groups.

Suggests subgrouping of age groups to make data more informative.

You may be divided into <60 and ≥60 or whatever, depending on your data.

We acknowledge that both quantitative and qualitative vaccine responses are impaired in elderly cohorts. However, in our small sample size, neither age as a continuous or categorical variable was associated with a difference in qualitative immune response. For this reason, we felt that subdividing each treatment group by age would add greater complexity to the visualization of the data, without aiding the findings of our trial.

Typos.

  1. "AstraZeneca" (no spacing between Astra and Zeneca).

This has been corrected

  1. Line 252: q-value.

Reviewer 3 Report

Comments and Suggestions for Authors

I was invited to revise the paper entitled "Dietary inulin to improve SARS-CoV-2 vaccine response in kidney transplant recipients: The RIVASTIM-inulin randomised controlled trial". It was a RCT aimed to evaluate efficacy of inulin supplementation to enhance the immune response to a third COVID-19 vaccine.

Observations:

- Sample size estimation was lacking. It is unknown how the sample size was estimated;

- Authors defined to enroll 120 patients but they reached only 94 patients;

- It is unclear if the analysis was performed as intention to treat;

- The main concern is regarding the previous covid-19 episodes. Authors did not evaluate the impact of previous diseases that can influence the titer;

- Among limitations, Authors did not reported the lack in sample size reaching.

Author Response

I was invited to revise the paper entitled "Dietary inulin to improve SARS-CoV-2 vaccine response in kidney transplant recipients: The RIVASTIM-inulin randomised controlled trial". It was a RCT aimed to evaluate efficacy of inulin supplementation to enhance the immune response to a third COVID-19 vaccine.

We thank the reviewer for their critique and analysis of our manuscript. The insightful feedback and required revisions have, in our opinion, strengthened the manuscript.

Observations:

- Sample size estimation was lacking. It is unknown how the sample size was estimated;

The justification for the target sample size has been previously published with the trial protocol. We have updated the manuscript to include the sample size justification.

“Whilst observational evidence supports the role of the gut microbiota in modulating the immune response to vaccination, the evidence supporting interventions which target the microbiota in this setting are lacking and estimates of effect size cannot be accurately generated.

We therefor considered the following when defining our sample size (1) the number of eligible KTRs across the two sites; (2) their current vaccination status; (3) the feasibility of conducting a trial within the contemporary resource setting; (4) the local prevalence of COVID-19 and (5) the recommended sample size requirements for a pilot study.(references) A recruitment target of approximately 60–120 participants was con-sidered feasible, and sufficient to generate a meaningful estimate of effect size with adequate confidence. With full recruitment of 120 participants, and assuming a 25% virus neutralisation endpoint in the control group, (references 4 and 13) we would require 54% virus neutralisation in the intervention group to demonstrate superiority using a one-sided hypothesis with 2.5% α-risk and 90% power.”

- Authors defined to enroll 120 patients but they reached only 94 patients;

This shortcoming is mentioned in the discussion

“Our trial is not without limitations. Firstly, the trial failed to reach its recruitment target and was hence underpowered to detect a difference in the primary endpoint. Recruitment began at the height of the COVI19 pandemic, and was hampered by the emergence of the Omicron variant. The resulting increase in both hospitalisations and public health restrictions led to a rapid upsurge in community access to vaccination, with many KTRs accessing a third vaccine outside of their transplant centre. For many others, delaying access to a third booster vaccine became ethically unacceptable.”

- It is unclear if the analysis was performed as intention to treat;

The primary analysis was performed as intention-to-treat (ITT), and is stated in the “Quantification and statistical analysis” section of the methods.

- The main concern is regarding the previous covid-19 episodes. Authors did not evaluate the impact of previous diseases that can influence the titer;

Patients were not eligible for enrolment if they had a documented prior infection with COVID-19, and this formed part of the published exclusion criteria. Throughout the trial period, participants were monitored and questioned closely for the development of adverse events, including infectious illness. In the placebo and inulin arms, 2 and 1 participant respectively, developed COVID and were removed from the primary analysis, as per the published protocol.

- Among limitations, Authors did not reported the lack in sample size reaching.

As mentioned above, the reasons for not achieving the target sample size are mentioned in the discussion.

Round 2

Reviewer 2 Report

Comments and Suggestions for Authors

That is okay to address the concerns.

However, refer to my major concern #2, which you already clarifying. 

Identifying an inert substance to use as a placebo for investigational microbiota trials remains a challenge, and one that was given much thought in our trial design. Maltodextrin is often used as a placebo due to its neutral taste, easy digestibility, and minimal impact on metabolic parameters at doses typically used in clinical trials. In our trial design, maltodextrin (placebo) was visually indistinguishable from the investigational product (inulin), and shared similar physical characteristics (solubility, taste), ensuring adequate masking of participants and investigators.

We acknowledge that some trials (largely in animal models) have identified maltodextrin as having a role in promoting intestinal inflammation through alterations in the mucous gel layer.1 However, major alterations in the composition of the microbiota have not been consistently observed2, and often not at the doses used in clinical trials. In human use, a recent metanalysis has suggested maltodextrin to impact human physiology and gut microbiota, however, the individual studies suggest considerable variability in the microbiota impacted, and the direction of the registered effect.3 Lastly, we believe the use of a placebo control arm was required to enhance study credibility, minimise bias, standardize care between groups, reduce protocol violations, and meet essential ethical considerations, and no other suitable alternative to maltodextrin is available.

  1. Laudisi F., Di Fusco D., Dinallo V., Stolfi C., Di Grazia A., Marafini I., Colantoni A., Ortenzi A., Alteri C., Guerrieri F., Mavilio M., Ceccherini-Silberstein F., Federici M., MacDonald T.T., Monteleone I., Monteleone G. The food additive maltodextrin promotes endoplasmic reticulum stress–driven mucus depletion and exacerbates intestinal inflammation. Cell Mol Gastroenterol Hepatol. 2019;7:457–473
  2. Arnold AR, Chassaing B. Maltodextrin, Modern Stressor of the Intestinal Environment. Cell Mol Gastroenterol Hepatol. 2019;7(2):475-476. doi: 10.1016/j.jcmgh.2018.09.014. Epub 2018 Oct 17. PMID: 30827413; PMCID: PMC6409436.
  3. Almutairi R, Basson AR, Wearsh P, Cominelli F, Rodriguez-Palacios A. Validity of food additive maltodextrin as placebo and effects on human gut physiology: systematic review of placebo-controlled clinical trials. Eur J Nutr. 2022 Sep;61(6):2853-2871. doi: 10.1007/s00394-022-02802-5. Epub 2022 Mar 1. Erratum in: Eur J Nutr. 2023 Aug;62(5):2345. PMID: 35230477; PMCID: PMC9835112."

I recommend you add a bit of your argument to the discussion to clarify the study's placebo used and make it sound and reasonable of your research design. I think other readers may raise this concern, just as I do.

Author Response

We thank the reviewer for identifying this oversite. Following the initial concern raised by the reviewer, the discussion should have been amended to include this discussion point: We have added the following text to the manuscript:

"Thirdly, using maltodextrin as an inert placebo presents certain issues. Maltodextrin is often chosen as a placebo in microbiota intervention trials due to its neutral taste, easy digestibility, and similar physical properties to the investigational product, as seen in this trial. While it was traditionally believed to have minimal impact on gut health and the colonic microbiota due to its rapid absorption in the small intestine, several studies suggest otherwise. In animal research, maltodextrin has been shown to contribute to intestinal inflammation by disrupting the gut mucous gel layer(1). Clinical studies in humans also indicate that maltodextrin can affect the gut microbiota's composition, though there is significant variability in the specific microbiota affected and the direction of these changes across different studies (2). Despite these concerns, no suitable alternative to maltodextrin has been identified or commonly used in microbiota trials. Omitting a control arm from the trial was considered, but it was determined that this would significantly undermine the study's scientific validity.

Reference 1.        Laudisi F., Di Fusco D., Dinallo V., Stolfi C., Di Grazia A., Marafini I., Colantoni A., Ortenzi A., Alteri C., Guerrieri F., Mavilio M., Ceccherini-Silberstein F., Federici M., MacDonald T.T., Monteleone I., Monteleone G. The food additive maltodextrin promotes endoplasmic reticulum stress–driven mucus depletion and exacerbates intestinal inflammation. Cell Mol Gastroenterol Hepatol. 2019;7:457–473

Reference: 2.        Almutairi R, Basson AR, Wearsh P, Cominelli F, Rodriguez-Palacios A. Validity of food additive maltodextrin as placebo and effects on human gut physiology: systematic review of placebo-controlled clinical trials. Eur J Nutr. 2022 Sep;61(6):2853-2871. doi: 10.1007/s00394-022-02802-5. Epub 2022 Mar 1. Erratum in: Eur J Nutr. 2023 Aug;62(5):2345. PMID: 35230477; PMCID: PMC9835112."

Reviewer 3 Report

Comments and Suggestions for Authors

It can be accepted 

Author Response

We thank the reviewer for their time, interest, and critic of our work, which have strengthened the manuscript.